# Specific Secondary Bile Acids Control Chicken Necrotic Enteritis

**DOI:** 10.3390/pathogens10081041

**Published:** 2021-08-17

**Authors:** Mohit Bansal, Tahrir Alenezi, Ying Fu, Ayidh Almansour, Hong Wang, Anamika Gupta, Rohana Liyanage, Danielle B. Graham, Billy M. Hargis, Xiaolun Sun

**Affiliations:** 1Center of Excellence for Poultry Science, University of Arkansas, Fayetteville, AR 72701, USA; mb043@uark.edu (M.B.); tjalenez@uark.edu (T.A.); yingfu@uark.edu (Y.F.); ama056@uark.edu (A.A.); hxw01@uark.edu (H.W.); ag048@uark.edu (A.G.); bmahaffe@uark.edu (D.B.G.); bhargis@uark.edu (B.M.H.); 2Cell and Molecular Biology (CEMB), University of Arkansas, Fayetteville, AR 72701, USA; 3Department of Chemistry, University of Arkansas, Fayetteville, AR 72701, USA; rliyana@uark.edu

**Keywords:** secondary bile acid, *Eimeria maxima*, *Clostridium perfringens*, intestinal inflammation, microbiota

## Abstract

Necrotic enteritis (NE), mainly induced by the pathogens of *Clostridium perfringens* and coccidia, causes huge economic losses with limited intervention options in the poultry industry. This study investigated the role of specific bile acids on NE development. Day-old broiler chicks were assigned to six groups: noninfected, NE, and NE with four bile diets of 0.32% chicken bile, 0.15% commercial ox bile, 0.15% lithocholic acid (LCA), or 0.15% deoxycholic acid (DCA). The birds were infected with *Eimeria maxima* at day 18 and *C. perfringens* at day 23 and 24. The infected birds developed clinical NE signs. The NE birds suffered severe ileitis with villus blunting, crypt hyperplasia, epithelial line disintegration, and massive immune cell infiltration, while DCA and LCA prevented the ileitis histopathology. NE induced severe body weight gain (BWG) loss, while only DCA prevented NE-induced BWG loss. Notably, DCA reduced the NE-induced inflammatory response and the colonization and invasion of *C. perfringens* compared to NE birds. Consistently, NE reduced the total bile acids in the ileal digesta, while dietary DCA and commercial bile restored it. Together, this study showed that DCA and LCA reduced NE histopathology, suggesting that secondary bile acids, but not total bile acid levels, play an essential role in controlling the enteritis.

## 1. Introduction

*Clostridium perfringens*, a spore-forming, anaerobic, and Gram-positive bacterium, is an opportunistic gut pathogen which causes necrotic enteritis (NE), a prevalent chicken disease [1,2,3]. NE is estimated to be responsible for losses of around USD six billion in the poultry industry worldwide every year [4]. The increased NE incidence is associated with the increasing restriction of prophylactic antimicrobial supplementation in farm animal production in the absence of effective alternatives [5]. Despite its largely elusive pathogenesis, NE at poultry farms is frequently intercurrent with the infections of coccidia such as *Eimeria maxima* and *E*. *acervuline* [6]. To mimic the enteritis in field conditions, the NE animal model is often a co-infection of *E. maxima* and *C. perfringens* [7]. Investigators have also successfully induced a natural NE by placing birds in rooms exposed to previous NE outbreaks [8,9]. Furthermore, researchers have used NE models by manipulating diets, microbiota, immune status, and gut homeostasis, such as a high fishmeal diet [10], a wheat-ray diet, immunosuppression, and others [11,12]. Among these NE models, the model of the co-infection of *Eimeria* and *C. perfringens* reproduces the clinical signs, mortality, and growth performance reduction of clinical and subclinical NE [3,8,13].

The main issue of NE in broiler production is the growth performance impairment on body weight gain (BWG) and feed efficiency. In severe clinical NE, birds suffer severe diarrhea and mortality, anorexia, and a reduced BWG and feed efficiency [13]. In subclinical NE, growth performance reduction is subtle and difficult to differentiate between healthy and NE birds. Besides a growth performance evaluation, NE, particularly severe NE, is often diagnosed through clinical signs and through the scoring of visible macro-lesions in the small intestine. The clinical signs of severe NE include sudden death, a decreased appetite, severe depression, watery to bloody (dark) diarrhea, closed eyes, and ruffled feathers. Scoring macro-lesions in the small intestine is based on intestinal thickness and friability, necrosis, hemorrhage, and gas production [13]. It is often difficult to discern the difference in subclinical NE with minimal gross necrotic lesions. A histopathology evaluation has been used in medical and veterinary diagnoses and research, and clinical and subclinical NE can be assessed with the microscopic histological method [1,7]. Scoring the microscopic histological lesions of NE evaluates the clear and subtle changes in the intestinal inflammation in the ileum tissue. This includes the scoring of the immune cell infiltration in the lamina propria and the submucosa, the fusion and blunting of villi, and the hyperplasia of crypts [1,7].

At the molecular level of pathogens, *C. perfringens* produces more than 16 protein toxins, which can cause histotoxic and severe intestinal inflammations in humans and animals [14]. *C. perfringens* expresses chromosomally or on plasmid alpha-toxins (a phospholipase C toxin, CPA) *C. perfringens* enterotoxin (CPE), necrotic enteritis B-like toxin (NetB), epsilon toxin, and beta toxin [15,16]. *C. perfringens* isolated from chickens always carries CPA which causes intestinal mucosal barrier dysfunction based on crude or partially-purified toxins [17]. Although NetB positive *C. perfringens* induces NE in certain experimental settings [4], the development of NE is not always associated with the presence of NetB [18,19]. In general, the uncertain role of *C. perfringens* toxins on NE development is clearly reflected in the current lack of an effective *C. perfringens* toxin vaccine against NE.

Despite the limited knowledge at the molecular level of the host, NE induces ileitis, and various inflammatory signaling pathways are activated, possibly comparable to other forms of enteritis, such as inflammatory bowel diseases. Adaptive and innate immunity is coupled with various signaling pathways that mediate intestinal inflammation such as the T helper (Th) cell type Th1, Th2, regulatory T (Treg), the Toll-like receptor (TLR), cyclooxygenase (COX), the mammalian target of rapamycin (mTOR), and the nuclear factor-κB (NF-κB) [20,21]. The signaling pathways induce the downstream expression of various proinflammatory mediators, such as IFNγ, IL17a, IL1β, IL8, and TNFα [22]. NE induces the gene expression of proinflammatory cytokines such as *Ifnγ* and *Il1β* in chicken intestinal tissue [1,13,23]. Ultimately, the inflammatory cytokines and chemokines mediate immune cell migration into the inflamed site, activate an inflammatory response, and cause cell death [24]. Cell death is categorized into accidental cell death and regulated cell death (RCD), and some of the RCDs are apoptosis, necrosis, necroptosis, and pyroptosis [25]. It remains largely elusive which death pathways are activated in NE, although we have reported that apoptosis is present in the enteritis [1]. 

Bile acids have long been investigated mainly in relation to the process of lipid digestion and absorption [26]. Bile acids are synthesized in the liver, secreted in the intestine, and metabolized by gut microbiota [27]. The bile acids include primary bile acids of a conjugated form of tauro- or glyco- cholic acids (TCA or GCA) and chenodeoxycholic acids (TCDCA or GCDCA), as well as deconjugated CA and CDCA by bacterial hydrolysis [28]. The deconjugated primary bile acids are metabolized by the intestinal microbiota into secondary bile acids of deoxycholic acid (DCA), lithocholic acid (LCA), and ursodeoxycholic acid UDCA [27]. Recently accumulated evidence has led to a re-examination of the key role of bile acids in infectious diseases, including *C. difficile* [29], NE [1], and campylobacteriosis [30]. 

When searching on PubMed with the term “poultry necrotic enteritis” on 15 February 2021, 105 articles were found from 1961 to 2000 (average 2.6 papers/year), while 472 papers were published from 2001 to 2020 (23.6 papers/year). The more than nine-fold increase in research papers between the two periods demonstrates the increasing interest and urgency in relation to the discovery of new NE interventions. In previous studies, we discovered that DCA prevented clinical and subclinical NE and restored the reduced ileal total bile acid level in NE birds [1,3]. We reasoned that the success of DCA could come from either increased total bile acids or elevated DCA alone. In this study, we aimed to address this reasoning through supplementing various bile acids to NE birds. The results suggest that specific secondary bile acids, but not total bile acids, are essential to prevent severe clinical NE. The findings will be important for designing new regiments against chicken NE and relevant enteritis.

## 2. Results

### 2.1. DCA and LCA Attenuated Clinical NE-Induced Ileitis

During the experiment, from day 23 to day 25 we observed that NE-challenged birds showed the typical clinical NE signs of a decreased appetite, severe depression, watery to bloody (dark) diarrhea, closed eyes, and ruffled feathers. Notably, the NE-challenged birds that were fed DCA or LCA diets showed reduced NE signs, while the NE-challenged birds that were fed chicken bile or commercial bile did not show much improvement. To evaluate the mechanism of DCA and LCA that was improving the NE signs, we then examined the impact of the dietary bile acids on NE at the cellular level using a histopathology analysis. The ileal samples were collected and processed for histology slides, and the histopathological score was evaluated as described in the method section. Consistent with the previous study [1], clinical NE-induced severe ileitis showed villus shortening, crypt hyperplasia, and massive immune cell infiltration in the lamina propria (Figure 1A). Remarkably, birds that were fed the DCA or LCA diets significantly reduced their ileitis and histopathological scores by 55 or 45% (6.0 and 7.3 vs. 13.1), respectively, compared to NE birds (Figure 1B). In contrast, dietary chicken bile or commercial bile failed to prevent NE-induced intestinal inflammation. The results suggest that NE reduction is dependent on the specific bile acids, but not on the total bile acid levels. 

### 2.2. DCA Attenuated Clinical NE-Induced Ileal Inflammation

Because the NE ileitis is associated with an intestinal pro-inflammatory response and the outcome of inflammatory response is cell stress and death [1], the TUNEL assay was used to characterize the effect of dietary bile supplementation on cellular apoptosis. Notably, severe NE-induced epithelial (arrows) and lamina propria immune cell death (arrowheads) was found in the ileal villi of the NE birds (Figure 2A). Consistent with the results of the histopathology evaluation, dietary DCA or LCA attenuated the cell apoptosis in the ileal villi, whereas chicken bile and commercial bile failed to reduce cell death.

We then reasoned that dietary DCA or LCA would reduce the NE-induced pro-inflammatory response. We then measured the host inflammatory mediator of mRNA expression in the bird ileal tissue using a real-time PCR. Consistent with the TUNEL results, the clinical NE induced a significantly higher accumulation of ileal inflammatory mRNA mediators of *Ifnγ* and *Mmp9* by 3.46- and 6.25-fold, respectively, compared to noninfected birds (Figure 2B). Dietary DCA reduced *Ifnγ* and *Mmp9* expression in the ileum tissue by 64% and 77%, respectively, compared to NE birds. Surprisingly, dietary LCA failed to reduce the proinflammatory mediators, although the bile reduced the NE histopathology and cell death. 

### 2.3. DCA Reduced C. difficils and E. maxima Colonization in the Ileum

It is reasonable to assume that the severity of the infectious intestinal disease is positively associated with the pathogen load. Hence, we reasoned that the reduction in the histopathology, inflammatory response, and cell death by DCA came from the decreased colonization of pathogens *E. maxima* and *C. perfringens*. We then quantified the pathogen colonization in the ileal luminal digesta using a real-time PCR. Notably, *C. perfringens* and *E. maxima* colonized at a 1.8-log (Figure 3A) and 7-log (Figure 3B) higher rate in the ileal lumen of NE birds compared to those of noninfected birds. Consistent with the previous report [3], dietary DCA reduced *C. perfringens* and *E. maxima* colonization by 28 and 25% in the ileum digesta, respectively, compared to NE birds, while dietary LCA, chicken bile, and commercial bile failed to reduce the pathogen colonization in ileal lumen. To further evaluate the spatial distribution of *C. perfringens*, fluorescence in situ hybridization (FISH) was performed on the histology tissue slides. Notably, *C. perfringens* in rod- or spore-shape (arrows) was present in ileal villus mucosa of the NE birds (Figure 3C), while the pathogen was barely visible in those of birds that were fed the DCA diet.

### 2.4. DCA Reduced Clinical NE-Induced BWG Loss

Because the important concern of poultry farmers during an NE outbreak is the reduction in the growth performance, we assessed the bird BWG during the various phases. As expected, birds grew comparably between different groups during the noninfected phase of day 0 to 14 and day 14–18 (Bile diets were started at day 14). Notably, during the coccidiosis phase of days 18–23, the BWG of the NE group of birds was reduced compared to that of noninfected control birds (33.51 vs. 51.13 g/d/bird, *p* < 0.05) (Figure 4). Interestingly, dietary bile supplementation did not prevent the BWG loss during the coccidiosis phase. Consistent with previous reports [1], birds infected with both *E. maxima* and *C. perfringens* developed severe ileitis during the NE phase of days 23–25 and suffered a loss of daily BWG compared to noninfected birds (−14 vs. 56 g/d/bird, *p* < 0.001). Notably, dietary DCA alleviated the NE-induced BWG loss (16 vs. −14 g/d/bird, *p* < 0.006) compared to NE birds, while dietary chicken bile, commercial bile, or LCA failed to attenuate BWG loss.

### 2.5. Dietary Bile Increased NE-Induced Total Bile Acid Level Reduction

From the results of the variable reduction in ileitis, inflammation, pathogen load, and BWG loss by the different dietary bile, we reasoned that it was possible the dietary bile failed to restore the total bile level in the ileum. Alternatively, it was possible that all the dietary bile increased the total bile pool in the ileum, but only specific bile could reduce NE. To address the possibilities, we collected ileal digesta from the experimental birds and quantified the bile acids using targeted metabolomics of bile. Consistent with the previous subclinical NE study [3], clinical NE significantly reduced the total bile acids by 66% in the ileum digesta of NE birds (2560 vs. 7524 nmol/g, *p* < 0.05) compared to the noninfected birds (Figure 5A). Notably, dietary commercial bile and DCA significantly increased the total bile acids in the ileum digesta compared to NE control birds (5378, 5964 vs. 2560 nmol/g), while dietary chicken bile and LCA did not significantly increase the NE-induced bile reduction. Interestingly, NE infection increased the ratio of unconjugated (CDCA and CA) to conjugated bile acids (T/GCDCA and T/GCA) in the ileum digesta compared to the noninfected group (2.73 vs. 0.25, *p* < 0.05) (Figure 5B), suggesting a possible increase in the deconjugating activity of bile salt hydrolase (BSH) or an increased conjugated bile acid absorption during NE. 

## 3. Discussion

We previously found that dietary DCA reduced severe clinical [1] and subclinical NE [3]. After quantifying the ileal bile acids in the subclinical study, we found that NE reduced the total ileal bile acids and DCA feeding restored the total bile level [3]. We then reasoned that the alleviation of NE by DCA was dependent either on the total bile acid level or the specific bile acid level (such as DCA) in the ileal digesta. In this study, we fed birds with chicken bile, commercial ox bile, DCA, or LCA to address this hypothesis. Consistently, NE reduced the total ileal bile acid level, while the DCA and ox bile diets restored the total ileal bile level comparable to that of noninfected birds. Notably, only the DCA diet was able to mitigate NE-induced intestinal inflammation, the inflammatory response, *C. perfringens* and *E. maxima* colonization, cell apoptosis, and BWG reduction. LCA only reduced NE-induced intestinal inflammation and cell death, while chicken bile and ox bile did not reduce NE pathology and BWG loss.

The notable finding in this study is that dietary bile supplementation was able to restore the ileal bile level reduced by NE. Consistent with the previous report [3], NE reduced the total bile acids in the ileal content by more than 66% compared to noninfected birds. The human bile pool is maintained relatively constant at about 3–5 g through two pathways: enterohepatic circulation and the de novo synthesis of bile acids. The majority of the bile pool (>95%) is conserved through enterohepatic circulation, whereas the latter pathway compensates for the daily fecal loss of bile acids [31]. The apical Na^+^-dependent bile salt transporter (ASBT) in the terminal ileum is responsible for the functional enterohepatic circulation [32]. Inflammatory bowel disease subjects with distal ileitis often suffer from bile acid malabsorption [33,34]. The expression of ASBT is reduced in ileal biopsies even from Crohn’s Disease patients free of any signs of inflammation [35]. The birds in this experiment had a severe NE at the ileum; hence, the loss of bile acids in the diarrhea excrete might have contributed to the reduced ileal bile pool. Although the two dietary bile supplements of DCA and commercial ox bile were able to compensate for the total ileal bile acid reduction, commercial ox bile failed to reduce NE histopathology, BWG loss, and other disease signs. Although dietary LCA did not restore the bile pool, the diet reduced NE-induced histopathology. Together, the results suggest that the specific secondary bile acid level, but not the total ileal bile acid level, is the driving force against NE histopathology.

In contrast to chicken and ox bile which are mainly primary bile acids, secondary bile acids DCA and LCA were able to reduce NE-induced ileal histopathology and cell apoptosis. The increasing body of knowledge indicates that different bile acids act differently on bacteria growth and disease induction. *C. perfringens* produces several toxins such as alpha (CPA), beta (CPB), epsilon (ETX), iota (ITX), enterotoxin (CPE), necrotic enteritis B-like toxin (NetB), and others [14]. The role of *C. perfringens* toxins on NE remains inconclusive, which is evident from the fact that there is no known toxin vaccine to effectively control NE [36]. Secondary bile acids DCA and LCA inhibit the in vitro growth and toxin production of *Clostridium difficile*, while primary bile acid CA fails to do so [29]. Because the chicken bile was mainly comprised of conjugated TCDCA and the ox bile was comprised of mainly G/TCA, the ileal bile of the birds that were fed the bile diet was mostly comprised of TCDCA/CDCA and TCA/CA, respectively. *C. perfringens* can transform conjugated primary bile acid to unconjugated primary bile acid [37,38], suggesting its potential resistance to primary bile acids. Secondary bile acid DCA reduces the in vitro vegetative growth of *C. perfringens* [39]. We previously reported that DCA, but not TCA and CA, reduced the growth of *C. perfringens* in vitro [1]. *C. perfringens* is a spore-forming bacterium, and its sporulation and CPE production are implicated in human enteritis [40]. We observed that DCA reduced *C. perfringens* spores in NE birds in a previous study [3], and it was not clear if the bacterial sporulation played any role in NE development. Interestingly, DCA reduced the culturable heat-resistant spore count by 6-logs, while CA only slightly (<1-log) reduced the culturable heat-resistant spore count [41]. Consistently, in current study, we observed that DCA reduced the vegetative and spore cells of *C. perfringens* in the ileal lumen and mucosa using DNA-based assays of qPCR and FISH. Because of the indistinguishability between live and dead bacteria by the DNA-based assays, caution should be taken toward the interpretation of the level of the pathogen colonization and invasion. Another secondary bile acid LCA mildly reduced the in vitro vegetative growth of *C. perfringens* [1]. However, dietary LCA induced a mouse liver injury with intrahepatic cholestasis and bile infarcts [42]. In this study, dietary LCA was able to reduce histopathology in NE birds, but it failed to reduce BWG loss. The toxicity of LCA to the liver might contribute to the failure of LCA against NE-induced BWG loss. In addition, because noninfected birds showed *C. perfringens* in the ileal digesta, the bacterium might have come from day-old chicks, feed, or the housing room. It remains unknown how much those “unintended” *C. perfringens* impacted the NE outcome. However, because NE birds and the birds with bile diets shared the same chick origin, feed, and room, the “unintended” *C. perfringens* would not impact our data interpretation. Together, the results suggest that specific bile acids, particularly DCA, protect birds against NE through actions that specifically antagonize the growth and possible sporulation and toxin production of *C. perfringens*.

Besides increasing evidence of the secondary bile acid DCA directly acting against the pathogen *C. perfringens*, the impact of bile acids on the host response has not been well studied. Although the evidence remains largely inconsistent on the role of specific adaptive immune responses in NE, an increased Th1 type IFNγ response is one of the consensuses in the NE model of co-infecting *Eimeria* and *C. perfringens* [12]. Consistent with this consensus, we found that the DCA diet significantly reduced NE-induced Th1 type *Ifnγ* gene expression in this study and previous clinical [1] and subclinical NE studies [3]. Other NE-induced proinflammatory mediators, such as *Il1β, Il17, Il23, Il22, Il8-1/2, Mmp9,* and *Litaf (Tnfα)*, were not consistently reduced by dietary DCA among different NE studies (partially unpublished data). Various factors could contribute to the differential results related to the inflammatory response in NE, as described in this review in detail [12]. For the role of the immune molecular signaling pathways on DCA and NE development, we reported that DCA prevented *C.*
*perfringens*-induced COX signaling in NE birds [1]. Inducible COX-2 activity mediates various inflammatory diseases, including inflammatory bowel disease [43] and radiation-induced small bowel injury [44]. COX-2 promotes gut barrier permeability and bacterial translocation across the intestinal barrier [45,46]. DCA might reduce NE-induced COX-2 signaling pathways. Interestingly, DCA attenuates *Campylobacter jejuni*-induced mouse colitis and mTOR signaling pathways in colonic tissues and immune cells [30]. mTOR, a downstream target of PI3K, has been implicated in many physiological activities, including cell growth, proliferation, survival, and innate and adaptive immune responses [47,48,49]. Further investigation will be necessary to identify whether *C. perfringens* modulates other pro-inflammatory signaling pathways, such as mTOR signaling pathways, in chicken NE. 

DCA and LCA are secondary bile acids derived from the metabolism of intestinal microbiota, which mainly inhabit chicken ceca. The microbiota population is relatively less dense in the small intestine, including in the ileum. It remains inconclusive how much the ileal or cecal microbiota directly influence the outcome of NE development, although the association of microbiota alteration and NE has been frequently reported in various papers which are reviewed in detail in this paper [12]. *C. difficile* infection in the human colon is clearly mediated by dysregulated microbiota, and microbiota transplantation prevents and treats the pathogen infection [29]. In contrast, no report is available on microbiota transplantation to prevent or treat NE in chickens. Unlike the risk factor of antibiotic pre-exposure in the *C. difficile* infection [29], prophylactic antibiotic supplementation protects birds against NE [5]. The spatial difference (colon vs. small intestine) between *C. difficile* and *C. perfringens* infection might play an important role in the pathogens’ infection alteration in response to microbiota presence. Future research on the role of microbiota on NE development is much needed.

Altogether, these data reveal that the presence of specific secondary bile acids was the key to reduce *C. perfringens*-induced NE. The total bile acid level alone was not sufficient to prevent NE in chickens. The reduction in NE by DCA reduced *C. perfringens* colonization, the inflammatory response, and cell death, ultimately leading to attenuated ileitis and BWG loss. These findings highlight the importance of understanding the molecular relationship between infectious pathogens, microbiota activities (e.g., bile acids), and inflammation. These discoveries could be useful for controlling NE and other intestinal diseases using microbiota and its metabolic products. 

## 4. Materials and Methods

### 4.1. Chicken Experiment

The chicken experiments were conducted at the Poultry Health Laboratory of the University of Arkansas at Fayetteville, Arkansas. Animal experiments performed were in accordance with the Animal Research: Reporting of In Vivo Experiments (https://www.nc3rs.org.uk/arrive-guidelines (accessed on 21 July 2021)). All experiments were approved by the Animal Care and Use Committee at the University of Arkansas. Day-old chicks obtained from Cobb-Vantress (Siloam Springs, AR) were wing-tagged and randomly allocated to individual cage pens. The birds were provided with their respective diet and water ad libitum. During the first five days, the temperature was maintained at 34 °C and then gradually lowered until a temperature of 23 °C was achieved on day 25. Corn–soybean meal-based starter diets and grower diets were fed from 0–10 and 11–25 days. The basal diet was formulated according to the broiler’s recommendations. Cohorts of birds included non-infected, NE, and NE infected birds fed with supplemented basal diets with 1.5 g/kg of commercial bile, LCA, or DCA (Alfa Aesar, MA, US), and 3.2 g/kg of chicken bile from day 14. No antibiotics, coccidiostats, or enzymes were added to the feed. There were 7 birds/pen and 2 pens/group except that LCA was 1 pen/group. On day 18, the birds were orally gavaged with 15,000 sporulated *E. maxima* M6 oocysts and on day 23 and 24 were orally infected with 10^9^ CFU *C. perfringens*/bird. This *C. perfringens* isolate was used in our previous studies [1,3] and was confirmed to be *cpa* and *netB* positive by PCR and proteomics. Day-old chicks and feed were not examined for the presence of *C. perfringens*. Chicken body weight was measured at day 0, 14, 18, 23, and 25. Five to ten birds from each group were sacrificed at day 25 to collect ileal tissue and digesta to analyze the histopathology, pathogen colonization, inflammation markers, and bile acid. The chicken bile was collected aseptically from the gall bladder of broilers at day 56 in the processing plant. The bile acid composition of the chicken bile was composed of 90% TCDCA, 9% TCA, and 1% other bile acids as determined by LC-MS/MS (Appendix A). The commercial ox bile was purchased from VWR (Ward’s Science, Rochester, NY, USA) and was composed of 45% GCA and 50% TCA as described by the manufacturer. DCA and LCA were purchased from VWR (all from Alfa Aesar). 

### 4.2. Histopathology Analysis of Intestinal Inflammation

The NE infection model of the co-infection of *Eimeria* and *C. perfringens* was developed by our collaborators and the lesions were evaluated using macroscopy evaluation [13]. For our studies, the NE lesions were assessed using the microscopic histopathology score system with the Swiss-rolling of the small intestinal region around Meckel’s diverticulum (about 8 cm down) and the macroscopy score was not able to be performed. The Swiss-roll was adopted from biomedical rodent research (sporadic colorectal cancer, colitis) and was able to capture sporadic and macroscopy-insensitive pathology. The Swiss-rolled ileal tissue samples were fixed in 10% phosphate-buffed formalin (pH 7.4) overnight at 4 °C. Samples of tissue were then embedded into paraffin blocks, and thin tissue sections (5 μm) were prepared. The tissue samples were processed and stained with H&E staining at the histology laboratory in the Department of Poultry Science at the University of Arkansas at Fayetteville. Using a Nikon TS2 fluorescent microscope, images were acquired. 

The histopathological score was assessed blindly. Briefly, each ileal tissue slide was divided into four sections, and each section was evaluated for inflammation based on the characteristic NE lesions as described before [3]. Total histopathological scores were then calculated by adding the four randomly selected area scores. A continuous (decimal) scoring method for intestinal lesions was used on a scale of 0–4 where 0 indicated: no inflammation, villi, and crypt intact; score 1: minor number of immune cells infiltration in villus laminar propria and submucosa, and little hyperplasia of crypts and blunting of villi; score 2: more extensive immune cells infiltration in villi laminar propria and submucosa and, villi shortened >1/4, or crypt hyperplasia; score 3: pronounced infiltration cells in the laminar propria of villi, crypts, submucosa, and muscularis, villi shortened >1/2 and edema, or crypt hyperplasia and regeneration; and score 4: necrosis, villus diffuse, ulcers, crypt abscesses, or transmural inflammation (may extend to the serosa).

### 4.3. C. perfringens and E. maxima Colonization in Ileal Lumen Using Real Time PCR and FISH

Ileal digesta samples from birds at day 25 were collected and weighed to extract DNA using the phenol/chloroform method as previously described [3]. A real-time PCR was performed to quantify the colonization level of *C. perfringens* and *E. maxima* in the ileal digesta targeting 16S and 18S rDNA, respectively. *C. perfringens* 16S rDNA was detected using a real-time PCR, and the primer sequences were: *16S*_forward: AGGAGCAATCCGCTATGAGA; *16S*_reverse: GTGCAATATTCCCCACTGCT and *E. maxima* (*Em*)18S_forward 5′-GACCTCGGTCACCGTATCAC-3′ *E. maxima* (*Em*)18S_reverse 5′-CGTGCAGCCCAGAACATCTA-3′. FISH was also performed to visualize *C. perfringens* intestinal colonization and invasion using histology slides as described before [3].

### 4.4. Host Inflammatory Response Using a Real Time PCR and a Terminal Deoxynucleotidyl Transferase dUTP Nick End Labeling (TUNEL) Assay

The NE-induced host immune response of inflammatory gene expression was quantified in ileal tissue samples. The total RNA was extracted by the TRIzol method from ileal tissue samples of birds at day 25 as described before [1,3] and cDNA was prepared using M-MLV (NE Biolabs). The accumulation of the proinflammatory genes of *Ifnγ*, *Mmp9*, and *Gapdh* in the ileum tissue was determined using SYBR Green PCR Master mix (Bio-Rad) on a Bio-Rad 384-well Real-Time PCR System as described before [1]. The gene expression of the fold-change was calculated using the ΔΔCt method [1,3] and *Gapdh* as an internal control.

The degree of ileal cells apoptosis in the intestinal tissue was assessed using the TUNEL assay as described before [1,3]. The TUNEL assay detected the late phase of cellular apoptosis based on the fragmentation of nucleic acids. In brief, ileal tissue slides were deparaffinized three times with xylene wash and then rehydrated with 100%, 95%, and 70% ethanol. The tissues were then incubated at 37 °C for 90 min with the TUNEL solution (5 μM Fluorescein-12-dUTP (Enzo Life Sciences), 10 μM dATP, 1 mM pH 7.6 Tris-HCl, 0.1 mM EDTA, 1U TdT enzyme (Promega)). For nucleus visualization, the slides were counter-stained utilizing DAPI. Using a Nikon TS2 fluorescent microscopy, the fluorescent green apoptotic cells were examined and imaged.

### 4.5. Quantification of Ileal Bile Acids Using Targeted Metabolomics

The ileal digesta were collected and the bile acid profile was analyzed using multiple reaction monitoring (MRM) mass spectrometry in the Statewide Mass Spectrometry Facility at the University of Arkansas in Fayetteville, Arkansas as described before [3]. Briefly, the ileal digesta samples from each group were used for the extraction of bile acids using the methanol method. To identify and quantify the bile acids, MRM methods were developed for the nine bile acids isotopically labeled (some were with deuteriums and some were with five deuteriums) and unlabeled standards as described before [3]. Calibration curves with internal standards were used to estimate bile acid concentration. LC-MS/MS analysis was performed in the negative-ion mode. LC separation was performed using a C18 column with flow rate of 0.3 mL/min. The three most intense multiple reaction monitoring (MRM) fragments for each bile acid standard were used as described before [3]. 

### 4.6. Statistical Analysis

Differences between the treatments were analyzed using a one-way ANOVA followed by a Fisher LSD multiple comparison test using Prism 7.0 software. Nonparametric data were also analyzed using the nonparametric Mann–Whitney U test using Prism 7.0 software. Values are shown as a mean of the samples in the treatment ± standard error of the mean as indicated. Experiments were considered statistically significant if the *p* values were <0.05.

## Figures and Tables

**Figure 1 pathogens-10-01041-f001:**
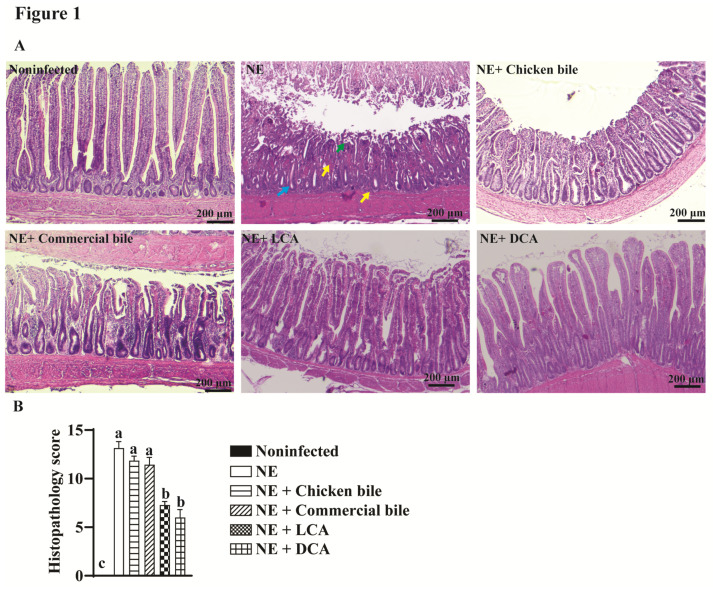
DCA Prevented Clinical NE-induced Ileal Inflammation. Cohorts of 7 to 14 broiler chickens were fed basal diets supplemented with different bile acids starting from day 14. Birds were challenged with *E. maxima* at day 18 and *C. perfringens* at day 23 and 24. The birds were sacrificed on day 25. (**A**) Representative images of H&E staining showing small intestinal histopathology. (**B**) Microscopic quantification of intestinal histopathological lesion score. Yellow arrows: immune cell infiltration; Green arrow: blunted villi; blue arrow: hyperplastic crypts. Different letters of a, b, and c mean *p* < 0.05. The scale bar is 200 μm. Results are representative of 3 independent experiments.

**Figure 2 pathogens-10-01041-f002:**
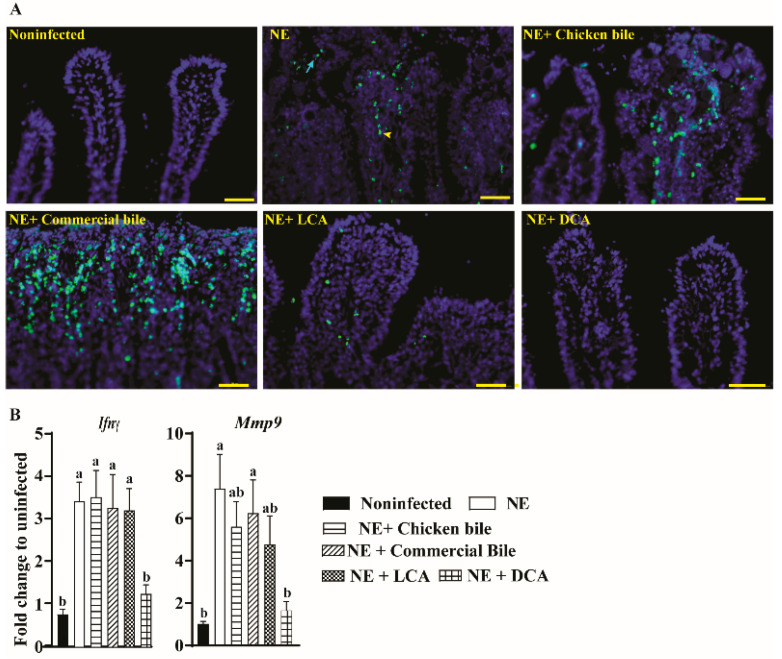
Cohorts of broiler chickens were fed different bile diets and infected as in Figure 1. (**A**) Representative images of the TUNEL assay showing cell death at the late stage of apoptosis (green dots). Yellow arrowhead: apoptotic immune cells in lamina propria; Green arrow: apoptotic epithelial cells. (**B**) Ileal *Ifnγ* and the *Mmp9* mRNA qPCR fold-change relative to noninfected birds and normalized to *Gapdh*. Different letters of a and b mean *p* < 0.05. The scale bar is 40 μm. Results are representative of 3 independent experiments.

**Figure 3 pathogens-10-01041-f003:**
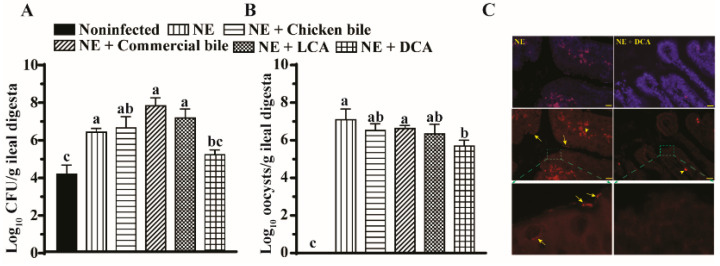
DCA reduced *C. perfringens* and *E. maxima* ileal colonization. Cohorts of broiler chickens were fed different bile diets and infected as in Figure 1. (**A**) Luminal *C. perfringens* colonization was quantified in the ileal digesta by measuring 16S rDNA using a qPCR. (**B**) Luminal *E. maxima* was quantified in the ileal digesta by measuring 18S rDNA using a qPCR. (**C**) *C. perfringens* mucosal invasion was visualized by FISH. Yellow arrows indicate the vegetative and spore cells (red) of *C. perfringens*. Yellow arrowheads indicate red blood cells (red). The blue color is the DAPI staining for the cell nucleus. The scale bar is 20 µm. All graphs show mean ± SEM. Different letters of a, b, and c mean *p* < 0.05. Results are representative of 3 independent experiments.

**Figure 4 pathogens-10-01041-f004:**
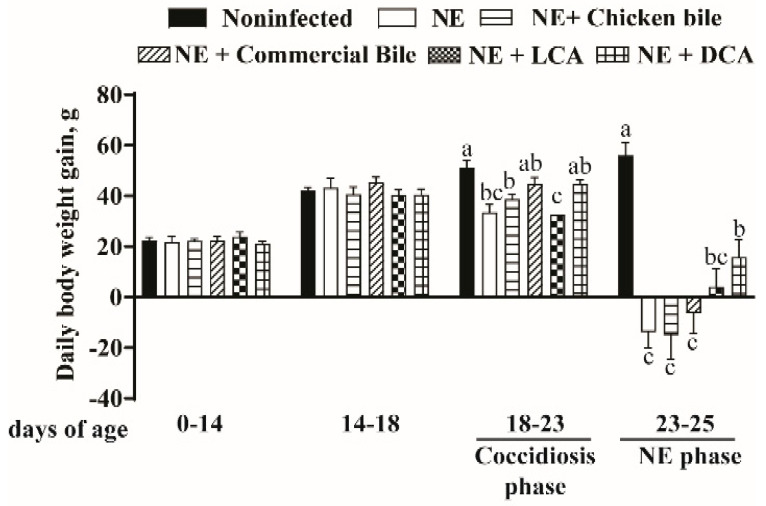
DCA reduced NE-induced productivity loss. Cohorts of broiler chickens were fed different bile supplemented diets from day 14 and infected as in Figure 1. Bird body weight gain (BWG) was measured at 0, 14, 18, and 25 days of age, and daily periodic BWG is shown. All graphs show mean  ±  SEM. Different letters of a, b, and c *mean p* < 0.05. Results are representative of 3 independent experiments.

**Figure 5 pathogens-10-01041-f005:**
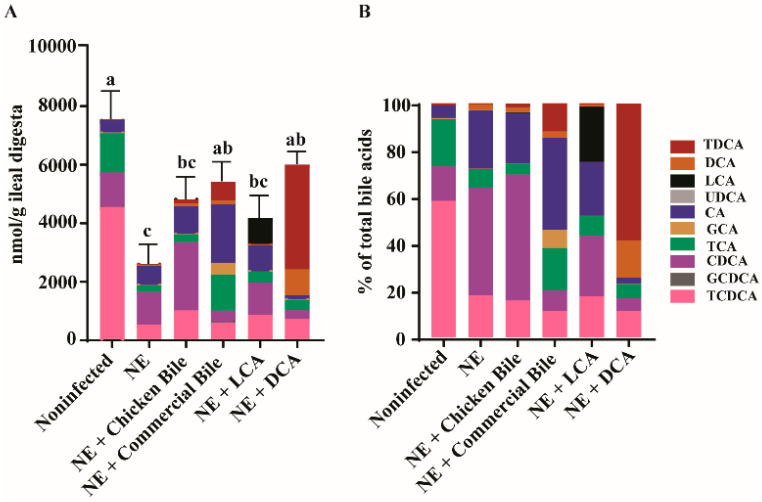
Dietary bile acids modulated the ileal bile acid composition in NE birds. Cohorts of broiler chickens fed with different bile acids and infected with NE as in Figure 1. (**A**) Total and individual bile acid quantification. (**B**) Relative composition of bile acids. All graphs show mean ± SEM. Different letters of a, b, and c mean *p* < 0.05. Results are representative of 3 independent experiments.

## Data Availability

Data are presented in this paper.

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
