# Peer review of "Specific Secondary Bile Acids Control Chicken Necrotic Enteritis"

_pathogens, 2021, doi:10.3390/pathogens10081041_

Round 1

Reviewer 1 Report

Overall, the manuscript is well written and the data collected support the conclusions of the authors. Data is presented with appropriate and descriptive presentation.

Statistics are appropriate.

The reviewer believes some attention to English/grammar of the written content is necessary.

A bit more discussion/explanation as to how DNA is related to the actual numbers of bacteria/coccidia are present in gut contents would be helpful to readers.

Author Response

Dear Editor:

We appreciate the comments from the editor and reviewers to improve this manuscript. We have added one Figure 3C and have made changes to the manuscript according to the reviewers’ comments. Most of the text changes were underlined. We hope that with these updates the manuscript will fit well with Pathogens’ standards. Below are the point-to-point responses to the reviewer’s comments. Thanks.

Sincerely,

Xiaolun Sun, PhD

Reviewer1:

Overall, the manuscript is well written and the data collected support the conclusions of the authors. Data is presented with appropriate and descriptive presentation.

Statistics are appropriate.

The reviewer believes some attention to English/grammar of the written content is necessary.

A bit more discussion/explanation as to how DNA is related to the actual numbers of bacteria/coccidia are present in gut contents would be helpful to readers.

Response: We appreciate the reviewer’s comments and we have made changes on the grammar of the written content. In addition, we have added the following in the Discussion section on topics of DNA and bacteria/coccidia at lines 292-296.

“Consistently, in current study, we observed that DCA reduced C. perfringens vegetative and spore cells in ileal lumen and mucosa using DNA-based assays of qPCR and FISH. Because of the indistinguishability between live and dead bacteria by the DNA-based assays, cautions should be taken toward the interpretation of the level of the pathogen colonization and invasion.”

Reviewer 2 Report

The manuscript ‘Specific secondary bile acids control chicken necrotic enteritis’ examined the impact of bile acids and C. perfringens induced necrotic enteritis.   This paper provided some interesting information, but in this reviewer’s opinion items in the context of data measurements and experimental design should be addressed.

Major comments:

Necrotic enteritis associated with C. perfringens infection can induce variable lesions within the gut and importantly measurements of microorganisms  within the enteric microbiota are often considered  to  determine the pathogenesis of disease. The authors did not measure changes in intestinal bacterial communities and this could affect the interpretation of any clinical finding. The authors need to discuss the potential  impact of intestinal microbial populations (ie colonization resistance etc) on the  enhancement  or reduction of disease following C. perfringens treatment.

Why were only luminal C. perfringens measured and not mucosal adherent C. perfringens as well. Mucosal  bound bacteria are often associated with the induction of disease.  Please address this in the discussion

Several pro -inflammatory cytokines were examined for intestinal injury. Although the investigators used  the TUNEL assay to determine cellular apoptosis, its still unclear why only these few cytokines were examined.  Often cytokine profiles representing the five  T-helper  effector systems (TH1, TH2, TH17, Treg, Tfollicular) as well as other (TNF-α, IL1 etc) are investigated. The authors should further elaborate  the reasoning for measuring these cytokines  and how other measured cytokines may  enhance the conclusion of the study.

Were the chicks tested (screened) for  C. perfringens after the chicks were received from the hatchery. Is the commercial hatchery free of C. perfringens.   If the chicks had pre-exposure to C. perfringens this could have a confounding affect on the data. Please address this in the methods or discussion.

Similar to the above comment- was the feed tested (screened)  for  C. perfringens contamination. If the feed was contaminated with C. perfringens this could have a confounding affect on the data Please address this in the method or discussion section

Minor comments:

Line 35: The reference for induction of necrotic enteritis  is quite old (2012). Other -newer methods for inducing disease (ie glucocorticoid treatment, Zaytsoff 2020 Gut Pathogens) have been developed. Elaborating on all methods for inducing disease should be considered within the introduction.

Line 328 : Were the individual(s) who assessed  histopathology ‘blinded’ to the treatment groups. Please address this  within the method section.

Line 112 should be  55%

Author Response

Dear Editor:

We appreciate the comments from the editor and reviewers to improve this manuscript. We have added one Figure 3C and have made changes to the manuscript according to the reviewers’ comments. Most of the text changes were underlined. We hope that with these updates the manuscript will fit well with Pathogens’ standards. Below are the point-to-point responses to the reviewer’s comments. Thanks.

Sincerely,

Xiaolun Sun, PhD

Reviewer2:

The manuscript ‘Specific secondary bile acids control chicken necrotic enteritis’ examined the impact of bile acids and C. perfringens induced necrotic enteritis.   This paper provided some interesting information, but in this reviewer’s opinion items in the context of data measurements and experimental design should be addressed.

Major comments:

Necrotic enteritis associated with C. perfringens infection can induce variable lesions within the gut and importantly measurements of microorganisms  within the enteric microbiota are often considered  to  determine the pathogenesis of disease. The authors did not measure changes in intestinal bacterial communities and this could affect the interpretation of any clinical finding. The authors need to discuss the potential  impact of intestinal microbial populations (ie colonization resistance etc) on the  enhancement  or reduction of disease following C. perfringens treatment.

Response: We appreciate the reviewer’s comments and we have added the following sentences in the Discussion section at lines 332-345?

“DCA and LCA are secondary bile acids derived from metabolism of intestinal microbiota, mainly inhabiting in chicken ceca. Microbiota population is relatively less dense in small intestine, including ileum. It remains inconclusive how much ileal or cecal microbiota directly influences the outcome of NE development, although the association of microbiota alteration and NE was frequently reported in various papers which are reviewed in detail in this paper [12]. C. difficile infection in human colon is clearly mediated by dysregulated microbiota, and microbiota transplantation prevents and treats the pathogens infection [29]. In contrast, no report is available on microbiota transplantation to prevent or treat NE in chickens. Unlike a risk factor of antibiotics pre-exposure in C. difficile infection [29], prophylactic antibiotics supplementation protects birds against NE [5]. The spatial difference (colon vs. small intestine) between C. difficile and C. perfringens infection might play an important role on the pathogens’ infection alteration in response to microbiota presence. Future research on the role of microbiota on NE development is much needed.”

Why were only luminal C. perfringens measured and not mucosal adherent C. perfringens as well. Mucosal  bound bacteria are often associated with the induction of disease.  Please address this in the discussion

Response: We appreciate the reviewer’s comments. We have added Figure 3C at page 6.

We added the following sentences in Result section at lines 149-153

“To further evaluate the spatial distribution of C. perfringens, fluorescence in situ hybridization (FISH) was performed on the histology tissue slides. Notably, C. perfringens in rod- or spore-shape (arrows) was present in ileal villus mucosa of the NE birds (Figure 3C), while the pathogen was barely visualizable in those of birds fed DCA diet.”

We also added the following sentence in the Method section at lines 418-419

“FISH was also performed to visualize C. perfringens intestinal colonization and invasion using histology slides as described before [3]”

Several pro -inflammatory cytokines were examined for intestinal injury. Although the investigators used  the TUNEL assay to determine cellular apoptosis, its still unclear why only these few cytokines were examined.  Often cytokine profiles representing the five  T-helper  effector systems (TH1, TH2, TH17, Treg, Tfollicular) as well as other (TNF-α, IL1 etc) are investigated. The authors should further elaborate  the reasoning for measuring these cytokines  and how other measured cytokines may  enhance the conclusion of the study.

Response: We appreciate the reviewer’s comments and we have added the following sentences in the Discussion section at lines 311-320.

“Although it remains largely inconsistent on the role of specific adaptive immune responses in NE, increased Th1 type IFNγ response is one of the consensuses in the NE model of co-infecting Eimeria and C. perfringens [12]. Consistent with this consent, we found that DCA diet significantly reduced NE-induced Th1 type Ifnγ gene expression in this study and previous clinical [1] and subclinical NE studies [3]. Other NE-induced proinflammatory mediators, such as Il1β, Il17, Il23, Il22, Il8-1/2, Mmp9, and Litaf (Tnfα), were not consistently reduced by dietary DCA among different NE studies (partially unpublished data). Various factors could contribute to the differential results of inflammatory response in NE, as described in this review in detail [12].  For the role of immune molecular signaling pathways on DCA and NE development, we”

Were the chicks tested (screened) for  C. perfringens after the chicks were received from the hatchery. Is the commercial hatchery free of C. perfringens.   If the chicks had pre-exposure to C. perfringens this could have a confounding affect on the data. Please address this in the methods or discussion.

Similar to the above comment- was the feed tested (screened)  for  C. perfringens contamination. If the feed was contaminated with C. perfringens this could have a confounding affect on the data Please address this in the method or discussion section

Response: We appreciate the reviewer’s comments and we have added the following sentences in the discussion at lines 301-305.

“In addition, because noninfected birds showed C. perfringens in ileal digesta, the bacterium might come from day-old chicks, feed, or housing room. It remains unknow how much impact of those “unintended” C. perfringens on NE outcome. However, because NE birds and the birds with bile diets shared the same chick origin, feed and room, the “unintended” C. perfringens wouldn’t impact our data interpretation.”

We have also added the following sentence in the method section at line 374.

“Day-old chicks and feed weren’t examined for the presence of C. perfringens.”

Minor comments:

Line 35: The reference for induction of necrotic enteritis  is quite old (2012). Other -newer methods for inducing disease (ie glucocorticoid treatment, Zaytsoff 2020 Gut Pathogens) have been developed. Elaborating on all methods for inducing disease should be considered within the introduction.

Response: We appreciate the reviewer’s comments and we have modified the following sentence at lines 34-36.

“Furthermore, researchers use NE models manipulating diets, microbiota, immune status, and gut homeostasis, such as high fishmeal diet [10], wheat-ray diet, immunosuppression, and others [11,12].”

Line 328 : Were the individual(s) who assessed  histopathology ‘blinded’ to the treatment groups. Please address this  within the method section.

Response: We appreciate the reviewer’s comments and we have added the following sentence in the method section at line 398.

“Histopathological score was assessed blindly.”

Line 112 should be 55%

Response: We apologize for the confusion. We have made changes at lines116-118.

“Remarkably, birds fed DCA or LCA diets were significantly reduced the ileitis and histopathological scores by 55 or 45% (6.0 and 7.3 vs. 13.1), respectively, compared to NE birds (Figure 1B).”

Round 2

Reviewer 2 Report

Thank you for the revised manuscript and addressing this reviewer’s  comments.